**Early Holocene ice on the Begguya plateau (Mt. Hunter, Alaska) revealed**
**by ice core [14]C age constraints**
Ling Fang[1,$], Theo M. Jenk[1,3*], Dominic Winski[4], Karl Kreutz[4], Hanna L. Brooks[4], Emma
Erwin[4], Erich Osterberg[5], Seth Campbell[4], Cameron Wake[6], Margit Schwikowski[1,2,3]
[1]Laboratory for Environmental Chemistry, Paul Scherrer Institute, CH-5232 Villigen PSI,
Switzerland
[2]Department of Chemistry and Biochemistry, University of Bern, CH-3012 Bern, Switzerland
[3]Oeschger Centre for Climate Change Research, University of Bern, CH-3012 Bern,
Switzerland
[4]Climate Change Institute and School of Earth and Climate Science, University of Maine,
Orono, Maine, 04469, USA.
[5]Deparment of Earth Sciences, Dartmouth College, Hanover, NH 03755
[6]Institute for the Study of Earth, Oceans, and Space, University of New Hampshire, Durham,
NH 03824
[$]Present address: Shaanxi Key Laboratory of Earth Surface System and Environmental
Carrying Capacity, Urban and Environmental Sciences department, Northwest University,
Xi'an 710127, China.
*Corresponding author

## Abstract

Investigating North Pacific climate variability during warm intervals prior to the Common Era can improve our understanding of the behavior of ocean-atmosphere teleconnections between low latitudes and the Arctic under future warming scenarios. However, most of the existing ice core records from the Alaska/Yukon region only allow access to climate information covering the last few centuries. Here we present a surface-to-bedrock age scale for a 210-meter long ice core recovered in 2013 from the summit plateau of Begguya (Mt. Hunter; Denali National Park, Central Alaska). Combining dating by annual layer counting with absolute dates from micro-radiocarbon dating, a continuous chronology for the entire ice core archive was established using an ice flow model. Calibrated $^{14}$C ages from the deepest section (209.1 m, 7.7 to 9.0 ka cal BP) indicate that basal ice on Begguya is at least of early Holocene origin. A series of samples from a shallower depth interval (199.8 to 206.6 m) were dated with near uniform $^{14}$C ages (3 to 5 ka cal BP). Our results suggest this may be related to an increase in annual net snow accumulation rates over this period following the Northern Hemisphere Holocene Climate Optimum (around 8 to 5 ka BP). With absolute dates constraining the timescale for the last > 8 ka BP, this paleo archive will allow future investigations of Holocene climate and the regional evolution of spatial and temporal changes in atmospheric circulation and hydroclimate in the North Pacific.

# 1 Introduction

Arctic surface temperatures have increased more than twice as fast as global temperature during the early 20th century and since the 1970s (Bengtsson 2004, Tokinaga et al. 2017, Svendsen et al. 2018). Recent modeling results suggest that during the early 20th century, as the Pacific Decadal Oscillation (PDO) transitioned to a positive phase, there was a concomitant deepening of the Aleutian Low that warmed the Arctic through poleward low-level advection of extratropical air (Svendsen et al. 2018). The impact of Pacific multi-decadal variability on Arctic warming has considerable implications for sea ice extent (Screen and Francis 2016), and hence the possible linkage between Arctic amplification, sea ice loss, and enhanced mid-latitude winter variability (Cohen et al. 2014, Francis et al. 2017, Cohen et al. 2018, Screen et al. 2018, Blackport et al. 2019, Cohen et al. 2019). Whether the present positive PDO conditions will persist and contribute to Arctic warming at an even higher rate in the future remains a fundamental question (Svendsen et al. 2018). A longer-term perspective on Pacific decadal variability and the teleconnection between the tropical Pacific, North Pacific, and the Arctic, particularly during warm intervals in the Holocene outside those captured in the instrumental record, would be an important contribution to this problem (e.g., Park et al. 2019). High-mountain ice cores in the North Pacific region have the advantage of sampling atmospheric moisture (e.g., snow), aerosol deposition, and preserving physical characteristics (e.g., melt), all of which can be related to Pacific climate processes (Zdanowicz et al. 2014, Osterberg et al. 2017, Winski et al. 2018), if Holocene (or greater) length records can be recovered.

The general timing of deglaciation in Alaska (Brooks Range, Central Alaska Range, and southern Alaska) was determined based on terrestrial cosmogenic radionuclides, lichenometry, and radiocarbon dating to between 10 and 20 ka BP (Dortch 2007). Following the Last Glacial Maximum (LGM), glaciers in the Brooks Range retreated up valley to, or even within, their modern limits by ca. 15 ka (Pendleton et al. 2015). Given the small extent of the Brooks Range glaciers prior to the Holocene thermal maximum, during which some glaciers in southern Alaska disappeared entirely (Barclay et al. 2009), it is possible that the Brooks Range glaciers may have disappeared as well. In the Central Alaska Range, reaching much higher altitudes and considering today's glacier extent, this is rather unlikely. Nevertheless, it is unclear where preserved ice from the early Holocene (or older) can be found in basal layers of these glaciers. Most of the ice cores recovered from the Alaska/Yukon region did not reach bedrock and are thus limited in the time covered, reaching back a few centuries only (Fig. 1). The Prospector

Russel Col (PRCol) ice core from Mt. Logan is an exception, having an estimated bottom age of ~20 ka BP based on the assumption that the significant depletion in the water stable isotope ratios observed in the very bottom section of the core is a signal of the LGM cold conditions (Fisher et al. 2008). The PRCol chronology is further constrained by a large $\delta^{18}O$ minimum and coeval increases in deuterium excess and $Ca^{2+}$ which are assigned to the 4.2 ka BP event (Walker et al. 2019), and tephra from the large Alaskan eruption of Aniakchak (3.6 ka BP, Walker et al. 2019). The PRCol record serves as a Global auxiliary stratotype for the Middle/Late Holocene subdivision boundary (Walker et al. 2019). However, there are no chronologic tie points in the PRCol record prior to the 4.2 ka BP event (Walker et al. 2019).

New surface-to-bedrock ice cores were recovered from the Begguya plateau (Mt. Hunter; Denali National Park, Alaska, 62.93°N/151.09°W; Fig. 1) in 2013 at 3900 m elevation (Winski et al. 2017). The two surface-to-bedrock cores (DEN-13A, DEN-13B) reached depths of 211.2 and 209.7 meters, respectively. Analysis of the upper 190 meters of DEN-13B (2013 to 810 CE) revealed that snow accumulation at the drilling site has doubled since ~1840 CE, coeval with warming of western tropical Pacific sea surface temperatures (Winski et al. 2017) and intensification of the Aleutian Low system (Osterberg et al. 2014, Osterberg et al. 2017). The same core also shows a sixty-fold increase in water equivalent of total annual melt between 1850 CE and present, which suggests a summer warming rate of $1.92 \pm 0.31$°C per century during the last 100 years in the altitude range of 3900 m (Winski et al. 2018). The Begguya melt layer record is significantly correlated with surface temperatures in the central tropical Pacific through a Rossby-wave like pattern that enhances temperatures over Alaska (Winski et al. 2018). Taken together, these hydroclimate changes are consistent with linkages between Pacific decadal variability and Arctic hydroclimate changes seen in the observational record (Svendsen et al. 2018), and demonstrate that the North Pacific hydroclimate response since 1850 CE is unprecedented in the past millennium.

The annual layer counting based chronology of the Denali core results in an ice age of $1203 \pm 41$ years at a depth of 190 m (152.8 m w.e.; Winski et al. 2017). Below that depth, annual layering was less consistent due to the loss of seasonal resolution caused by the glacier flow-induced thinning of layers. However, based on previously reported depth-age scales of ice cores from cold, high-elevation glaciers frozen to bedrock, the bottom 20 meters of ice may contain most of the record in terms of time, covering the Holocene and potentially even reaching into the Last Glacial (Uglietti et al. 2016, Licciulli et al. 2020). The Denali ice core therefore provides the possibility of establishing a new Holocene North Pacific hydroclimate

record reaching beyond the Common Era, if a precise and absolutely-dated chronology can be established in the bottom 20 meters of the core. The water-insoluble organic carbon (WIOC) and dissolved organic carbon (DOC) $^{14}$C-dating method has been validated and applied for multiple mid-latitude ice cores (e.g. Jenk et al. 2009, Uglietti et al. 2016, Hou et al. 2018, Fang et al. 2021). The technique makes use of the transport and deposition of carbonaceous aerosols onto the glacier. Before the industrial period, carbonaceous aerosols were mainly emitted from the living biosphere and from biomass burning. Consequently, this carbon reflects the contemporary atmospheric $^{14}$C content (Jenk et al. 2006). After deposition, the WIOC and DOC is incorporated into glacier snow, firn, and ice and undergoes radioactive decay with a half-life of 5730 years (Godwin 1962). Here we report results from $^{14}$C analysis of the bottom 60 m of the Denali ice core. These absolute dates extend the existing late Holocene Begguya chronology (Winski et al. 2017), providing the first high latitude Northern Hemisphere ice core chronology based on absolute dates from radiometric methods. We discuss our results in relation to Holocene ice extent and climate in the North Pacific region.

## 2 Methods

### 2.1 Annual layer counting

Two surface-to-bedrock ice cores (DEN-13A, DEN-13B) were drilled in 2013 at 3,900 meters elevation above sea level (a.s.l.) from the saddle between the north and middle peaks of Begguya (Mt. Hunter), Alaska (Winski et al. 2017; Osterberg et al. 2017; Winski et al. 2018; Polashenski et al. 2018). The annual layer counting for DEN-13B was previously published (Winski et al. 2017) and is only briefly described here. The timescale from 2013 to 1777 CE was determined by counting annual oscillations in $\delta^{18}$O (summer peak), melt layers (summer peak), magnesium (spring peak), dust (spring peak), liquid conductivity (summer peak), ammonium (summer peak) and methanesulfonic acid (MSA; late summer-fall peak), consistent with previous North Pacific ice cores (Yasunari et al. 2007, Osterberg et al. 2014, Tsushima et al. 2015). Between 1777 to 1500 CE annual layer counting is based on annual oscillations of $\delta^{18}$O, $\delta$D, dust concentration and liquid conductivity that were measured at higher resolution than the other analytes, while conductivity and dust concentrations were exclusively used to date the ice core from 1500 back to 810 CE. For this study, the counting based on these two parameters has been extended back in time (see section 3.2).

## 2.2 Denali ice core $^{14}$C analysis

Sixteen samples were selected from the lower portion of the DEN-13B (Table 1). Because WIOC concentrations at this site were assumed to be low, ice samples of at least 1 kg of mass were cut, aiming for extracted yields of carbon allowing dating with a reasonable uncertainty of 10-20% (> 10 µg C, Uglietti et al. 2016). In order to process such large sample volumes, a splitting of the sample for melting was required and the overall filtration time had to be increased. Using artificial ice produced from ultra-pure water, the adapted procedures were tested to reach low blanks similar to the ones previously achieved for smaller samples (Jenk et al., 2009; Uglietti et al., 2016; Fang et al., 2019). Otherwise, the samples for WIOC $^{14}$C-dating were prepared following the protocol described in Uglietti et al. (2016) with a brief summary provided here. In order to remove potential contamination in the outer layer of the ice core, pre-cut samples from the inner part of the core were rinsed with ultra-pure water. After melting of the sample in a pre-cleaned jar (1L, PETG, Semadeni), due to the size split in two, the carbonaceous particles contained as impurities in the sample ice were filtered onto a prebaked quartz fiber filter (Pallflex Tissueqtz-2500QAT-UP). Potential particulate carbonates also remaining on the filter were removed by acidifying three times with 0.5 µL of 0.2 M HCl. These initial steps were performed in a class 100 laminar flow box to ensure clean conditions. At the University of Bern (Laboratory for the Analysis of Radiocarbon with AMS-LARA laboratory) the WIOC samples were then combusted in a thermo-optical OC/EC analyzer (Sunset Modeldoc4L, Sunset Laboratory Inc, USA) with a non-dispersive infrared sensor for $CO_2$ quantification, using the established Swiss 4S protocol for OC/EC separation (Zhang et al. 2012). Being coupled to a 200 kV compact accelerator mass spectrometer (AMS, mini radiocarbon dating system MICADAS), equipped with a gas ion source and a Gas Interface System (GIS, Ruff et al. 2007, Synal et al. 2007, Szidat et al. 2014), the LARA Sunset-GIS-AMS system (Agrios et al. 2015, Agrios et al. 2017) allowed for final, online $^{14}$C measurements of the $CO_2$ produced from the WIOC fraction.

For the deepest sample from ~209 m depth (Denali235) the available amount of ice was very limited (~200 g). To ensure sufficient mass of carbon for final AMS analysis, the $^{14}$C dating was performed on the DOC fraction for which a higher concentration compared to the WIOC fraction is expected (Legrand et al. 2013). By a catalyzed UV-Oxidation in a dedicated system, DOC was converted to $CO_2$ which was then cryogenically trapped and flame sealed in glass ampules for final AMS analysis. Details can be found in Fang et al. (2019).

All [14]C results are expressed as fraction modern ($F^{14}C$), which is the $^{14}C/^{12}C$ ratio of the sample divided by the same ratio of the modern standard referenced to the year 1950 CE (NIST standard oxalic acid II, SRM 4990C) both being normalized to -25‰ in $\delta^{13}C$ to account for isotopic fractionation. Daily AMS calibration was performed using sets of modern (NIST oxalic acid II, SRM 4990C, $F^{14}C = 1.3407 \pm 0.0005$) and fossil standards (sodium acetate, Sigma-Aldrich, No. 71180, $F^{14}C = 0.0018 \pm 0.0005$). Final values presented in Table 1 are the AMS $F^{14}C$ raw data after corrections accounting for constant contamination and cross contamination in the Sunset-GIS-AMS system (or GIS-AMS system for DOC, respectively) and the overall procedural blank contribution introduced from preparation of ice samples to final AMS analysis. $F^{14}C$ of DOC was corrected for contribution from $^{14}C$ in-situ production following Fang et al. (2021). The applied small shift in $F^{14}C$ of $0.019 \pm 0.010$ was derived using an in-situ production rate of 260.9 $^{14}C$ atoms $g_{ice}^{-1}$ $a^{-1}$ as the best estimate for the site latitude and elevation (Lal et al. 1987, Lal and Jull 1990, Lal 1992), an average accumulation rate of $1.0 \pm 0.5$ m w.e. (a best initial guess based on the annual values from Winski et al. 2017, ranging from 0.2 to 2.0 m w.e. for the time period 810 to 2013 CE), and assuming an average incorporation into DOC of $18 \pm 7\%$ (Hoffmann, 2016). This correction shifts the calibrated age by $300 \pm 200$ years older, with uncertainty being fully propagated as for all other ages. Note that the upper estimate does not exceed the achieved dating precision defined by the analytical uncertainty (see Table S1 in the Supplementary). For all samples, calibrated radiocarbon ages were derived by calibrating final $F^{14}C$ values using OxCal v4.4.4 (Ramsey 2021) with IntCal20 (the Northern Hemisphere calibration curve; Reimer et al. 2020) and the OxCal in-built sequence model (Bayesian approach-based deposition model; Ramsey 2008, Ramsey 2017). All calibrated $^{14}C$ ages are presented as the 1σ range in years before present (cal BP, with BP referring to the year 1950 CE).

## 3 Results

### 3.1 Englacial stratigraphy

Around the Begguya drill site, no folding was observed in ground penetrating radar (GPR) data and the bedrock geometry appears to be uncomplicated (Campbell et al. 2013). New radar data was collected in 2022. Ice thickness, bed topography, and internal stratigraphy of the core site were mapped using GPR (10 MHz center frequency radar system, Blue Systems Integration). Standard processing techniques were applied to the data: clipping stationary periods, applying horizontal stacking, bandpass filtering, and correction for antenna separation (Lilien et al.

2020). Data were interpolated for standard trace spacing and then migrated using the SeisUnix sumigtk routine. Clear, visible layering is evident in the majority of the ice column; however, interpretation of the stratigraphy at depth is complicated by sidewall reflections produced from the trough beneath the ice core site. There is no conclusive evidence from this data of either stratigraphic continuity or discontinuity in the bottom-most 10 m of ice (Fig. 2). Future measurements using the millimeter-precision capabilities of autonomous phase sensitive radar (Brennan et al. 2014) would be beneficial to resolve englacial stratigraphy close to the bedrock.

### 3.2 Annual layer counting

Annual layer counting (ALC), previously published in Winski et al. 2017 back to 810 CE (section 2.1.), was extended back to 339 CE, i.e. for the top 197 meter. The uncertainty in the ALC chronology back to 810 CE was estimated through statistical comparisons among individual layer positions indicated by three individuals (see Winski et al. 2017 for details). By 1900 CE, uncertainty estimates are $\pm$ 4 years, increasing to $\pm$ 10 years at 1500 CE and $\pm$ 30 years by 810 CE (190.05 m). Only one individual (DW) performed ALC below 190 m, prohibiting a similar approach to estimate uncertainties, but we estimate an uncertainty of around $\pm$ 60 years at 339 CE. These estimates are for ALC only and do not consider additional, constraining information from time horizons. There is no offset between the timescale and inferred volcanic eruptions as indicated by peaks in sulfate, chloride, and conductivity during the 19[th] and 20[th] centuries, indicating that an accuracy within $\pm$ 2 year throughout the last 200 years is likely. The sulfate and chloride peaks in the 18[th] century used for chronology validation (inferred as Laki, 1784 CE and Pavlof, 1763 CE) were offset by one year from the ALC chronology. Additionally, $^{137}$Cs concentrations in the Denali core strongly peak in the layer assigned to the year 1963 CE, one year after the most extensive atmospheric testing of nuclear weapons, which matches the $^{137}$Cs residence time in the atmosphere (Tian et al., 2007, Winski et al., 2017).

### 3.3 Denali ice core $^{14}$C data

Air masses leading to precipitation on Begguya (~3900 m asl.) originate predominantly from the Pacific and contain relatively low organic aerosol concentrations (Haque et al. 2016, Choi et al. 2017). The WIOC concentration in the Denali core is thus significantly lower than in ice cores from the Alps. The WIOC concentrations range from 6 to 31 $\mu$g C kg$^{-1}$ ice with an average of 13 $\pm$ 7 $\mu$g C kg$^{-1}$ (Table 1). This is slightly higher than in snow at Summit, Greenland (4.6

µg C kg$^{-1}$, Hagler et al. 2007), but only about half of the pre-industrial WIOC concentrations
in European Alpine ice cores, with 24 ± 9 µg C kg$^{-1}$ (Legrand et al. 2007) and 32 ± 18 µg C
kg$^{-1}$ (Jenk et al. 2009) from Colle Gnifetti, Monte Rosa, Switzerland and 24 ± 7 µg C kg$^{-1}$ from
Fiescherhorn Glacier (Jenk et al. 2006). In agreement with findings from previous studies
(Legrand et al. 2007), the concentration of DOC (80 µg C kg$^{-1}$), measured in the deepest sample,
was significantly higher than the concentration of WIOC.

236         $^{14}$C calibration was performed using the OxCal in-built sequence model (Ramsey, 2008,

Ramsey 2017; see *Methods*). The assumption that samples are in chronological order allows
statistical constraints for the most likely age distribution of the individual samples in the
sequence. This assumption of chronological ordering will be discussed below. Samples
containing less than 10 µg C are generally characterized by a wide range of age probability. A
reduction in the dating precision for those samples is expected due to the small carbon amount
available for analysis. Small amounts on the one hand cause reduced AMS measurement
precision (lower $^{12}$C current and less $^{14}$C counts) and a lower, unfavorable signal-to-noise ratio
(i.e. the ratio between size of sample and procedural blank) on the other hand. Combined, this
leads to a larger overall analytical uncertainty, finally translating into a wider range of possible
ages. Although we used a considerable amount of ice for each sample (~1 kg), the total carbon
amount in 5 samples was significantly below this 10 µg C threshold recommended to obtain a
reliable dating with a final uncertainty < 20% for samples older than around 1000 years
(Uglietti et al., 2016). These samples will thus not be discussed in the following (but can be
found in the supplement material, together with calibration results without sequence constraint).

251         Calibrated $^{14}$C ages range from 0.3 ± 0.3 ka cal BP at 166.2 m (131.4 m w.e.) depth to

8.4 ± 0.6 ka cal BP for the deepest sample (Denali235; 209.1 m or 169.8 m w.e.), the last
sample above bedrock (0.6 m). These results show the characteristic exponential increase in
age with depth, expected for a cold glacier archive due to the associated ice flow dynamics (e.g.
Dansgaard and Johnsen, 1969, also see section 4.1.), and most importantly, reveal ice of early
Holocene origin in the Denali ice core (Table 1 and Figure 3). The absolute ages from
radiocarbon dating are in agreement with the independently derived ages from the ACL
reported in Winski et al (2017), extended back to 339 CE in this study (see *Annual layer*
*counting*). For the youngest sample, Denali183 from a depth of 166.2 m (131.4 m w.e.), and
for Denali214 from 192.6 m (155.0 m w.e.), the 1σ age range is 4–679 a cal BP and 958–1410
a cal BP, respectively; similar to the corresponding ACL derived ages of 340–380 and 1200–
1410 a BP. The 1σ $^{14}$C age range for Denali210-211 at 189.5 m (152.3 m w.e.) is 527–930 a
cal BP and with a possible age of 930 a cal BP only slightly younger than the ACL derived age
range of 1020–1200 a BP (in agreement within the 2σ range of 317–1174 a cal BP).
Samples of indistinguishable ages, with regard to the achieved dating uncertainty (i.e.
analytical precision), were observed in the depth interval from 200.3 to 206.2 m (161.9 to 167.2
m w.e.). This interval corresponds to a time period from around 3.2 to 4.3 ka BP. For the
respective samples (Denali223, Denali224-225, Denali229-230, Denali231), a low Agreement
Index (denoted as $A$ in OxCal) resulted for the applied $^{14}$C calibration approach. $A$ indicates the
level of agreement between the probability function derived by the ordinary calibration
approach (a priori distribution) and the calibration with additional constraint (a posterior
distribution; see OxCal and Ramsey, 2008 and 2017 for more details). Distributions are shown
in Figure 3. A value of 100 indicates no alteration in the distribution (100% or unity) while a
value lower than 60 indicates a warning to check for the validity of the underlying assumption,
i.e. (i) a non-sequential layering of samples, or (ii) the presence of analytical outliers. It is
apparent from Figure 3, that the two samples with lowest $A$ (<10), Denali223 and 231, are also
characterized by an exceptionally large uncertainty. For the batch of samples with AMS Lab
ID BE-10013.1.1 to BE-10022.1.1 (Table 1; see also Supplement Figure S1 and Table S1), the
contribution to the final overall uncertainty from AMS analysis only was around twice as much
than what typically can be achieved for samples of that carbon mass. For that measurement
day, we also observed above average uncertainties for the measured sets of AMS calibration
standards, with a slight elevation in the fossil standard value (+0.02 in F$^{14}$C; see *Method*). This
is an indication for non-ideal AMS conditions due to sub-optimal instrument tuning on the one
hand, and an elevated, potentially non-stable background that day on the other hand. Thus,
neither $^{14}$C ages nor the englacial stratigraphy give sufficient evidence to conclude a non-
sequential ordering of samples (i.e. an age reversal in the Denali ice core). Additionally, there
is evidence from other studies from the region suggesting hydrological changes between
around 4 to 2 ka BP (e.g. increased lake levels and precipitation, see *Discussion*), which
coincides with the time period in question here. Because increased accumulation rates would
lead to a reduced increase in age per unit depth, an unambiguous resolving of the sequence then
depends on the achievable analytical uncertainty. Having pushed the limit of the analytical
method with the small amounts of carbon available for $^{14}$C analysis and considering all the
above, we thus exclude assumption (i) and are confident that the applied $^{14}$C sequence
calibration approach does provide us the most accurate dates.

## 4 Discussion

### 4.1 Denali ice core chronology

Modeling the age scale in high-elevation mountain ice cores can be attempted either by applying rather simple glaciological one-dimensional (1D) flow models (e.g. Nye 1963, Dansgaard and Johnsen 1969, Bolzan 1985) or by much more complex 3D models based on a suit of observational data from glaciological survey (e.g. Campbell et al. 2013, Licciulli et al. 2020). Independent of model complexity, age scale modeling, particularly of mountain glaciers, is strongly challenged to provide accurate or even conclusive ages along the profile at a specific point on the glacier (e.g. the ice core drill site; Campbell et al. 2013, Licciulli et al. 2020). This is especially the case close to bedrock, where ice flow can become highly complex, and because past annual net accumulation rates with potential variations over time are unknown. Layers of known age along a glacier depth profile, e.g. from ice core dating, can provide crucial model constraints, allowing free model parameters to be tuned for a best fit between observations and model output. For a defined point, moving along a single axis (bed to surface), 1D models benefit from their simplicity to do so (less parameters). 1D models have been applied for decades to obtain continuous age-depth relationships at sites on polar ice sheets (e.g. Dansgaard and Johnsen 1969), thereby also accounting for past changes in accumulation rates by inverse modelling approaches (e.g. Buiron et al. 2011, Buchardt and Dahl-Jensen 2008). However, applications to sites from high-mountain glaciers are more recent (e.g. Jenk et al. 2009, Uglietti et al. 2016).

In the case of the Denali ice core, accurate dating by ACL supported with independent time horizons for the upper two thirds of the core and absolute dated horizons for the deep section of the core ($^{14}$C dates) are available. Winski et al. (2017) developed a well-defined age scale for the upper part of the core based on ACL supported by distinct time horizons. Since depth-age relationships are less challenging to model in the upper 90% of the ice core, because of relatively moderate layer thinning and little if any influence from bedrock, Winski et al. (2017) used a combination of 1D modeling and a 3D glacier flow model developed for this site (Campbell et al. 2013) to determine a significant increase in accumulation rates since around 1850 CE. Therefore, significant changes in net accumulation rates at the Denali ice core drill site should be expected to a have also occurred in the more distant past.

Due to its simplicity, we used the 1D two-parameter model (2p-model; Bolzan 1985) to
provide a first, best estimate for a continuous age-depth relationship from surface to bedrock,
building on the available data points presented. The 2p-model is based on a simple analytical
expression for the decrease of the annual layer thickness with depth and has two degrees of
freedom, the mean annual net accumulation rate $b$ and the thinning parameter $p$, characterizing
the strain rate function; both assumed to be constant over time. Knowing the glacier thickness
of 209.7 m from the ice core length (supported by ground penetrating radar data; 170.4 m w.e.)
and with all depths converted from meter to meter water equivalent based on the ice core
density profile, allowed finding the best solutions for $b$ and $p$ by fitting the model (least squares
approach, as described in Fahnestock et al. 2001) through the time horizons in the Denali ice
core (Volcanos, $^{137}$Cs, $^{14}$C). The derived value for $p$ was $0.79 \pm 0.01$. The resulting value of $b$
of $1.5 \pm 0.1$ m w.e. yr$^{-1}$, representing the mean annual net accumulation rate for the entire period
covered by the ice core, is similar to the recently observed 21$^{st}$ century values. It is however
significantly higher than the average value of around 0.5 m w.e. yr$^{-1}$ previously determined for
the last 810 years (ranging from around 0.3 to 1.5 m w.e. yr$^{-1}$; Winski et al. 2017). This is no
surprise, considering the likelihood that similar variations may also have occurred further back
in time. As a consequence of being constrained by the age of dated layers, the model results
are in agreement with the observational data for the total time period covered within the ice
column. However, at various depths along the depth profile, a significant offset between model
output and data can be observed (Fig. 4a). Again, this is not unexpected, considering the fact
that the accumulation rate was kept constant in the model, while significant changes over time
are known to have occurred (Winski et al. 2017). In Figure 3a, the effect on model results for
variations of $b$ is illustrated (runs with b equal to 0.5, 1.5 and 2 m w.e. yr$^{-1}$, respectively, with
$p$ as determined before).
To achieve our final goal, obtaining a continuous age-depth relationship based on the
absolute dating presented, we next applied a simple inverse modeling approach. We tightly fit
the model to the experimental data, by numerically solving for the exact value of $b$ for each
depth with a determined age ($p$ and $H$ as before). To reduce and account for potential noise in
the data, an uncertainty weighted three point running mean to obtain the non-steady state values
for $b$ was calculated (starting from top, then reversed from bottom, thereby propagating the
values for continuity). These values, interpolated for depths between the dated layers, were
finally used for model input, yielding a continuous age-depth relationship (Figure 4b and 4c).
All uncertainties have been fully propagated throughout calculations (from analysis to
modeling). We derived annual net accumulation rates of $0.5 \pm 0.1$ m w.e. yr$^{-1}$ at around 1000
CE, eventually increasing to a 20$^{th}$ century average value of $1.1 \pm 0.2$ m w.e. yr$^{-1}$ (Fig. S2).
This is in good agreement with what was determined previously by Winski et al. 2017 for the
corresponding periods, based on results from different models investigated (for the 3D model
considered best: 0.25 m w.e. yr$^{-1}$ around 1000 CE, with models ranging from 0.05–0.7 m w.e.
yr$^{-1}$, and $1.1 \pm 0.3$ m w.e. yr$^{-1}$ for the 20$^{th}$ century average, respectively). During the Holocene
Climate Optimum (around 8 to 5 ka BP, Kaufman et al. 2016) we obtained net accumulation
rates of $1.2 \pm 0.3$ m w.e. yr$^{-1}$, similar to the average rate observed since 1950 CE, followed by
higher rates of $1.7 \pm 0.4$ m w.e. yr$^{-1}$ from around 4.3 to 3.2 ka BP. Then, the rates decrease over
the next 500 to 1000 years to around $0.4 \pm 0.2$ m w.e. yr$^{-1}$. See Section 4.3 for further discussion.
Our derived age-depth scale results in ages of 9–14 ka BP at 0.5 m above bedrock, strongly
suggesting the presence of, at least, early Holocene ice at the Denali ice core drill site.
**4.2 Ice core chronologies in Eastern Beringia**
So far, existing ice cores from Eastern Beringia (Table 2) were dated with ages covering less
than the last millennium except for the Denali core discussed in this study and the 188 m long
PRCol core (Fig. 1) drilled to the bed surface on the summit plateau of Mt. Logan in 2001 and
2002. The older part of the PRCol core was dated based on a signal interpreted as the Younger
Dryas to Holocene transition (sudden reduction in electrical conductivity coinciding with a
drop in $\delta^{18}O$ and an increase in various chemical species) and a bottom age estimate from an
ice flow model of about 20 ka (Fisher et al. 2008). Another ice core from Mt. Logan (King Col,
60.59$^{o}$N,140.60$^{o}$W, 4135 m a.s.l.) was drilled in 2002 reaching a depth of 220.5 m. This core
was not dated, but a potential age range of 0.5 to 1.3 ka was estimated based on modeling
results (Shiraiwa et al. 2003). The 152 m ice core drilled in 2008 on the McCall Glacier was
dated by using a combination of ACL and specific horizons. The upper 37 m of ice date back
to 65 years and the full 152 m core was estimated to cover more than 200 years but no actual
dating of the lower section was performed (Klein et al. 2016). The Aurora Peak site is located
southeast of Mt. Hayes and the ice core was also drilled in 2008. The total ice thickness at the
drilling site is $252 \pm 10$ m, but this core (180.17 m) did not approach the bed surface. By annual
layer counting, the estimated bottom age of the Aurora Peak core is about 274 years (Tsushima
et al. 2015). Two cores were collected at Eclipse Icefield in 2002. The chronology of these
cores is based on multi-parameter ACL of seasonal oscillations in the stable isotope ($\delta^{18}O$) and
major ion records (Na$^{+}$) supported by identification of volcanic horizons. The longest core 2
(345 m) covers the period 1000 CE to 2002 CE (Yalcin et al. 2007), but did not reach bedrock.
In 2004, a 212 m ice core was drilled from Mt. Wrangell. The ice depth in the summit caldera
is probably over 900 m, but the definite bottom has not yet been detected (Benson et al. 2007).
For this core, a short 12-year record of dust and $\delta$D was reported in Yasunari et al. (2007), and
dating was later extended back to 1981 (23 years) at the depth of 100.1 m (Sasaki et al. 2016).
The record from Mt. Waddington only covers a period of 1973–2010 CE (Neff et al. 2012).
The total length of the Mt. Waddington core is 141 m, but the total ice thickness at the drilling
site is about 250 m. The ice core from Bona-Churchill reached bedrock at a depth of 460 m,
but the age-depth scale has only been established for the last ~800 years (depth of 399 m); the
deepest ice is estimated to exceed 1500 years in age (Porter et al. 2019).
Because none of the cores from the Eastern Beringia region was either drilled to the bed
surface or the ice close to the bed dated by an absolute dating method, no concluding evidence
about the age of the oldest glacier ice preserved in this region existed so far. In this study, we
achieved a first, complete and absolute (radiometric) dating by a first application of $^{14}$C analysis
on a high-latitude Northern Hemisphere ice core from Begguya, which reached the to bed
surface. Our results, with calibrated $^{14}$C ages of 7.7 to 9.0 ka BP close to the bottom (0.61 m
above bedrock) and model based indication for potentially even older ice further below (>12
ka BP), clearly indicate that glaciers in this region can be of early Holocene or even Pleistocene
origin.
**4.3 Possible implications for Holocene hydroclimate in Eastern Beringia**
In recent decades, extensive work has been done in the North Pacific region to characterize
Holocene hydroclimate (Table 3, Fig. 5). Following a modest Holocene thermal maximum that
was 0.2–0.5° C warmer than the last millennium average (Kaufman et al. 2016), although 1.7°
C cooler than present (Porter et al. 2019), glaciers across the region advanced synchronously
at about 4.5 ka BP (Solomina et al. 2015). This Neoglaciation continued through 3.5 to 2.5 ka
BP in the Yukon Territory based on past tree line variations, lake levels and carbonate oxygen
isotopes (Denton and Karlén 1977, Anderson et al. 2005a). While a mid-Holocene temperature
decrease may have played a role, Denton and Karlen (1977) hypothesized that an increase in
regional precipitation contributed to the regional Neoglaciation, a conclusion also reached by
later studies (e.g. Anderson et al., 2011).
Concurrent with this Neoglaciation, effective moisture rose across much of the region.
Based on pollen reconstructions, Heusser et al. (1985) inferred a doubling of Southern Alaskan
mean annual precipitation from around 3.9 to 3.5 ka BP (Fig. 3). Clegg and Hu (2010) found
that effective moisture, particularly during winters, increased markedly between 4.0 and 2.5 ka
BP. Hansen and Engstrom (1996) suggested cooler and wetter conditions in Glacier Bay at
around 3.4 ka BP. At Jellybean Lake and Marcella Lake, lake levels were high between 2.0–
4.0 ka BP (Anderson et al. 2005a, 2005b) which was assigned to changes in the strength and
positions of the Aleutian Low (Anderson et al. 2005b), consistent with the more recent
interpretation of hydroclimate changes from the Denali ice core (Winski et al. 2017; Osterberg
et al. 2017). Records from Mica Lake (Schiff et al. 2009) and Sunken Island Lake (Broadman
et al. 2020) showed wetter conditions associated with a stronger Aleutian Low at 4 ka and 4.5
ka BP, respectively. Greenpepper Lake experienced high lake levels from 2–5 ka BP (L.
Anderson et al. 2019) and a major shift from moss to sedge occurred at Horse Trail Fen
concurrent with a large isotopic anomaly at 3–4 ka BP (Jones et al. 2019). At the same time,
paleoenvironmental records showed a decrease in wildfire (Anderson et al. 2006; Kelly et al.

435   2013).

436       Together, previous work indicates an enhanced flux of moisture into the region, likely

associated with a strengthened Aleutian Low, sometime near 4 ka BP (Anderson et al. 2016).
The Denali ice core may provide corroborating evidence for this mid-Holocene shift in
hydroclimate (Table 3, Fig. 5). As presented before, samples of indistinguishable ages, at least
for the achieved analytical precision, were observed in the depth interval from 200.3 to 206.2
m (161.9–167.2 m w.e.) corresponding to the modeled time period from $4.3 \pm 0.5$ to $3.2 \pm 0.5$
ka BP (see Sections *Denali ice core $^{14}C$ data* and *Denali ice core chronology*). Elevated snow
accumulation provides a possible explanation for this clustering of dates and would support
many previous studies. While our model results based on $^{14}$C ages are consistent with existing
interpretations of mid-Holocene changes in regional precipitation, applying other independent
dating methods using the remaining parallel ice sections from this depth interval (e.g. from
DEN-13B) could be used, and additional geophysical and modeling approaches are needed to
rigorously test this hypothesis.

449       Importantly, some hydroclimate studies do not show a shift toward wetter conditions at 4

ka BP. On Adak Island, conditions grew cooler and drier at 4.5 ka BP which is consistent with
the prevailing interpretation of a stronger Aleutian Low advecting warmer moister air into the
Gulf of Alaska and cooler drier air to the western Aleutians (Bailey et al. 2018).  Certain sites
located to the north of the Alaska or St. Elias ranges also show a drying trend or no major
features around 4 ka BP (Lasher et al. 2021; Finney et al. 2012; King et al. 2022; Chakraborty
et al. 2010), emphasizing the idea that orography and rain shadows are critical for controlling
the relationship between site precipitation and circulation (Anderson et al. 2016).  In fact,
Winski et al. (2017), showed that during the instrumental era, Begguya snowfall is highly
correlated with precipitation along the Gulf of Alaska, but bears little resemblance to nearby
precipitation recorded in interior Alaska; a pattern that seems to hold through the mid-Holocene.
We note that the Aleutian Low is a wintertime phenomenon such that the role of summertime
precipitation may be an important contributor to some of the observed variability among
regional paleorecords. Comparing records with different seasonality or with seasonal
resolution will be critical in the future given that most of the isotope-based records listed above
are dominated by wintertime Aleutian Low dynamics.

## 5 Conclusion

Although [14]C analysis of ice-incorporated carbonaceous aerosols has allowed radiocarbon
dating of various high-elevation ice cores from low- and mid-latitudes, this technique has not
been applied before for high latitude ice cores because of the generally lower carbon content.
The [14]C results from the Denali ice core are the first from a high latitude ice core. These were
achieved by small adaptations in the ice sample preparation procedures for the WIOC [14]C-
dating method which allowed processing of larger ice samples up to >1 kg of ice and the
application of a new technique for [14]C dating of the DOC fraction, which benefits from higher
concentration levels in ice compared to the WIOC fraction (by around a factor of three).
Combining dating by ACL to a depth of 197.2 m (159.2 m w.e.; ~1674 years BP or 339 CE,
respectively), volcanic tie-points from sulfate, chloride, conductivity, and the new [14]C dated
horizons, a complete continuous chronology over the entire core was established using a simple
inverse ice flow modeling approach. For the overlapping sections, ages based on ACL are
confirmed by the agreement with the absolute, radiometric [14]C dates.

480       [14]C dating of a sample from just 0.61 m above bedrock at around 209 m depth, yielded

the first absolute date for near-bedrock ice in the region. Dated to be 7.7 to 9.0 thousand years
old, our result clearly indicates this very bottom ice to be of early Holocene age. The additional
model results indicate a high likelihood of even older ice below (>12 ka). The old ice at the
bottom of the Denali core confirms that at least some glacier ice in the central Alaskan Range
survived the Holocene thermal maximum. Future, independent dating methods would be

beneficial to further constrain and improve the timescale presented here. Our results show the applicability and great potential of $^{14}$C dating on low carbon content samples from North Pacific/Arctic ice cores. While they indicate the Denali ice core to currently be one of the few existing archives in the North Pacific region providing an opportunity to reconstruct Holocene hydroclimate variability, we do expect that similar or even longer paleo ice core records can be recovered from North Pacific glaciers if bedrock can be reached.

**Data availability.** All $^{14}$C data are available in the supplementary material.

**Supplementary material.** Additional Figures and Tables for this article can be found in the Supplementary.

**Author contributions**. LF, TMJ and MS performed $^{14}$C analysis, evaluation, and the continuous age-depth scale modeling, DW, KK, EO, SC, HLB and CW drilled the core and/or conducted the chemical and physical properties analysis. HLB, DW, and EO identified the annual layers. EE provided the radar image. LF, TMJ, DW, KK and MS wrote the manuscript while all authors contributed to the discussion of the results.

**Competing interests**. There is no conflict of interest.

**Special issue statement.** This article is part of the special issue "Ice core science at the three poles (CP/TC inter-journal SI)".

**Acknowledgements**. Thanks go to the Laboratory for the Analysis of Radiocarbon with AMS (LARA) at the University of Bern, especially to Martin Rauber for his help with operating the Sunset-MICADAS system. We thank Denali National Park, Polar Field Services and Talkeetna Air Taxi for providing air support and field assistance, Mike Waszkiewicz for ice core drilling, Brad Markle, Dave Silverstone, Tim Godaire and Elizabeth Burakowski for field assistance, and to more than 25 students for their support in the field and the lab. The work in this manuscript was funded by the U.S. National Science Foundation (AGS-1806422 and AGS-AGS-2002483). FL was supported by State Key Laboratory of Cryospheric Science, Northwest Institute of Eco-Environment and Resources, Chinese Academy Sciences (Grant Number: SKLCS-OP-2021-02).

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

**Table 1** $^{14}$C results of the Denali ice core samples (DEN-13B), given as F$^{14}$C, $^{14}$C ages, and calibrated $^{14}$C ages. For $^{14}$C calibration, chronological layering was assumed (sequential deposition, see main text). Samples were dated using the WIOC fraction, except for section 235 in which the DOC fraction was analysed. Numbers of the carbon amount available for $^{14}$C AMS analysis as well as the concentration of WIOC (DOC) in the sample are also provided. Additionally shown is the range of the dating based on ALC (range from top to bottom depth of section) and the final age scale (inverse ice flow model).

| Sample ID | AMS Lab ID | Depth (m) | Mid Depth (m w.e.) | Carbon amount (µg C) | WIOC (µg kg⁻¹) | F$^{14}$C (1σ) | $^{14}$C age (a BP, 1σ) | Calibrated $^{14}$C age (a cal BP, 1σ range) | Final age scale (a BP) | ALC (a BP) |
|---|---|---|---|---|---|---|---|---|---|---|
| Denali164 | BE-10013.1.1 | 148.6–149.4 | 115.90 | 7 | 6.2 | 0.910 ± 0.058 | 758 ± 513 | -[*] | 160–180 | 150–180 |
| Denali183 | BE-10015.1.1 | 165.7–166.6 | 131.40 | 11 | 10.1 | 0.921 ± 0.042 | 661 ± 367 | 4–679 | 350–370 | 340–380 |
| Denali209 | BE-10016.1.1 | 187.8–188.7 | 151.16 | 9 | 9.8 | 0.826 ± 0.044 | 1536 ± 428 | -[*] | 1010–1060 | 980–1090 |
| Denali210-211 | BE-8997.1.1 | 188.7–190.3 | 152.29 | 11 | 20.0 | 0.922 ± 0.033 | 652 ± 288 | 527–930 | 1080–1130 | 1030–1190 |
| Denali214 | BE-10017.1.1 | 192.1–192.9 | 155.00 | 14 | 11.8 | 0.831 ± 0.036 | 1487 ± 348 | 958–1410 | 1160–1420 | 1230–1380 |
| Denali215-216 | BE-8998.1.1 | 193.0–194.7 | 156.17 | 9 | 12.0 | 0.925 ± 0.039 | 626 ± 339 | -[*] | 1200–1560 | 1290–1500 |
| Denali217 | BE-10018.1.1 | 194.7–195.5 | 157.33 | 7 | 6.1 | 0.731 ± 0.054 | 2517 ± 594 | -[*] | 1280–1710 | 1400–1560 |
| Denali219-220 | BE-8615.1.1 | 196.4–197.3 | 159.31 | 12 | 16.8 | 0.841 ± 0.026 | 1391 ± 248 | 1242–1706 | 1560–1970 | >1420 |
| Denali223 | BE-10019.1.1 | 199.8–200.7 | 161.93 | 21 | 17.3 | 0.608 ± 0.029 | 3997 ± 383 | 3079–3469 | 2180–2890 | - |
| Denali224-225 | BE-11923.1.1 | 200.7–202.3 | 163.06 | 34 | 17.5 | 0.653 ± 0.010 | 3423 ± 123 | 3257–3530 | 2470–3510 | - |
| Denali228 | BE-10020.1.1 | 203.5–204.2 | 165.11 | 9 | 10.0 | 0.627 ± 0.043 | 3750 ± 552 | -[*] | 2860–3850 | - |
| Denali229-230 | BE-11924.1.1 | 204.2–205.7 | 166.09 | 39 | 20.0 | 0.691 ± 0.009 | 2969 ± 105 | 3305–3566 | 3040–4040 | - |
| Denali231 | BE-10021.1.1 | 205.7–206.6 | 167.18 | 11 | 11.5 | 0.523 ± 0.037 | 5207 ± 569 | 3840–4263 | 3540–4560 | - |
| Denali232-233 | BE-11925.1.1 | 206.6–208.1 | 168.26 | 55 | 30.8 | 0.629 ± 0.008 | 3724 ± 102 | 4067–4407 | 4520–5430 | - |
| Denali234 | BE-10022.1.1 | 208.1–208.8 | 169.23 | 10 | 11.7 | 0.378 ± 0.043 | 7814 ± 918 | 7264–8406 | 6270–9650 | - |
| Denali235[#] | BE-12465.1.1 | 208.8–209.4 | 169.83 | 21 | 80.3$_{DOC}$ | (0.437 ± 0.025) 0.418 ± 0.027[$] | 6649 ± 447 7007 ± 520 | 7737–8987[$] | 8920–13140 | - |

[*]Following recommendations, samples with a carbon mass of significantly less than 10 µg C were not considered (Uglietti et al. 2016).

[#]Results from the DOC fraction.

[$]After correction for in-situ $^{14}$C production (Fang et al. 2021; see main text).

**Table 2** Overview of existing North Pacific ice cores.

| Site | Year of drilling (CE) | Latitude (ºN) | Longitude (ºW) | Elevation (m a.s.l.) | Depth (m) | Reported time span (a) |
|---|---|---|---|---|---|---|
| McCall Glacier[a] | 2008 | 69.17 | 143.47 | 2310 | 152 | >200 |
| Aurora Peak[b] | 2008 | 63.52 | 146.54 | 2825 | 180 | 274 |
| Begguya[c] | 2013 | 62.56 | 151.05 | 3900 | 208 | >8,000 |
| Mt. Wrangell[d] | 2004 | 62.00 | 144.00 | 4317 | 212 | 23 |
| Bona-Churchill[e] | 2002 | 61.40 | 141.42 | 4420 | 461 | ~800 |
| Mt. Logan PRCol[f] | 2001-2002 | 60.59 | 140.50 | 5340 | 188 | ~20,000 |
| Eclipse Icefield[g] | 2002 | 60.51 | 139.47 | 3017 | 345 | ~1,000 |
| Mt. Waddington[h] | 2010 | 51.38 | 125.26 | 3000 | 141 | ~40 |


[a]McCall Glacier (Klein et al. 2016), [b]Aurora Peak (Tsushima 2015), [c]Begguya (this study), [d]Mt.
Wrangell (Yasunari et al. 2007; Sasaki et al., 2016), [e]Bona-Churchill (Porter et al. 2019), [f]Mt. Logan
(Fisher et al. 2008), [g]Eclipse Icefiled (Yalcin et al. 2007),[h]Mt. Waddington (Neff et al. 2012)

**Table 3** Regional paleoclimate events.

| Location | Reference | Paleoclimate events | Time (ka BP) |
|---|---|---|---|
| Begguya | This study | Elevated net accumulation rates | 4.3 ± 0.5 to 3.2 ± 0.5 |
| Yukon Territory | Denton and Karlén 1977; Anderson et al. 2005b | Neoglaciation | 3.5 to 2.5 |
| St. Elias Mountains | Denton and Karlén 1977 | Glacier extension | 3.6 to 3.0 |
| Alaska | Solomina et al. 2015 | Glacier extension | 3.5 to 3.0 |
| Marcella Lake | Anderson et al. 2005b | High lake levels | 4.0 to 2.0 |
| Greenpepper Lake | Anderson et al. 2019 | High lake levels | 5.0 to 2.0 |
| Jellybean Lake | Anderson et al. 2005a | Intensified Aleutian Low | 4.0 to 2.0 |
| Mica Lake | Schiff et al. 2009 | Intensified Aleutian Low | 4.0 ± 0.5 |
| Sunken Island Lake | Broadman et al. 2020 | Intensified Aleutian Low | 5.0 to 4.0 |
| Takahula Lake | Clegg and Hu 2010 | High effective moisture | 4.0 to 2.5 |
| Horse Trail Fen | Jones et al 2019 | Isotopic anomaly | 4.0 to 3.0 |
| Southern Alaskan | Heusser et al. 1985 | Precipitation increases | 3.9 to 3.5 |
| Kenai Lowlands | R.S. Anderson et al. 2006 | Decrease in wildfire | 5.5 to 4.5 |
| Yukon Flats | Kelly et al. 2013 | Decrease in wildfire | 5.0 to 4.0 |


799 .

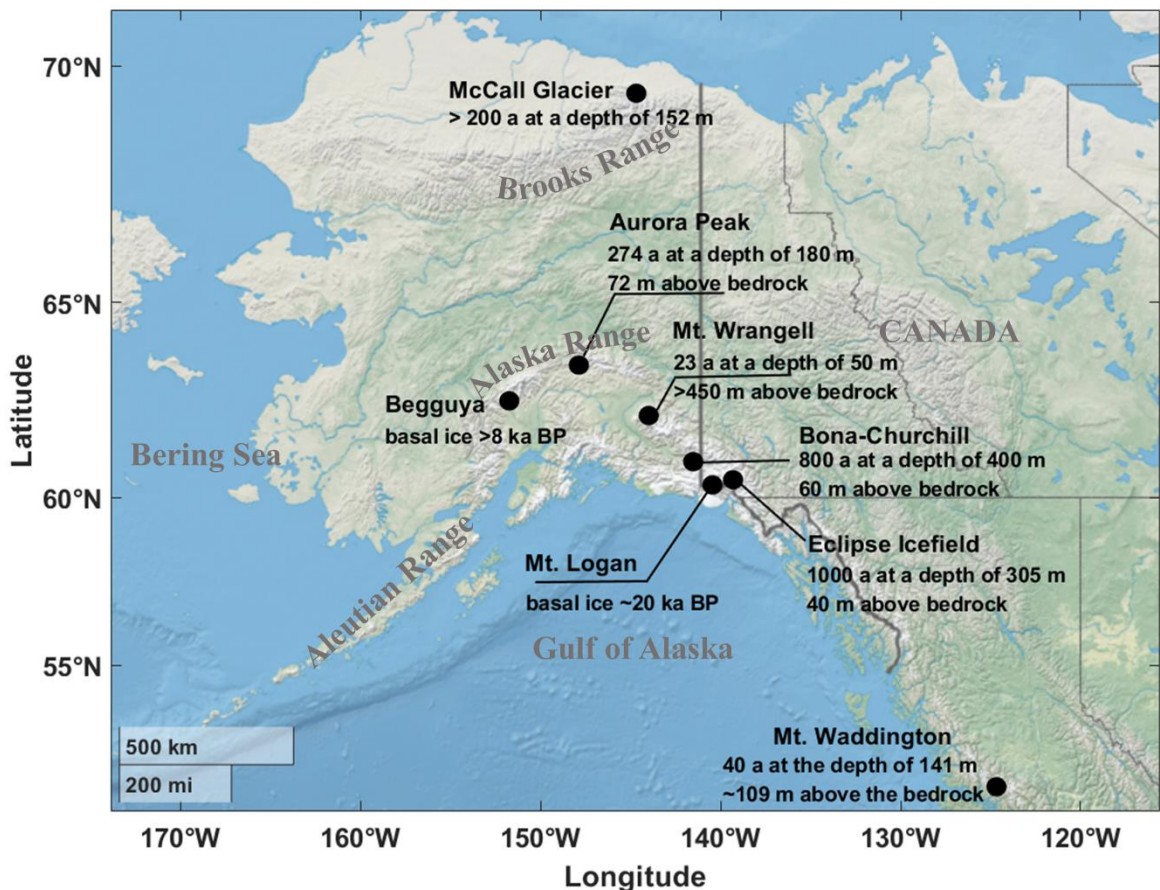

**Figure 1** Location map of North Pacific ice core sites and the age of the oldest ice dated from each location: Begguya (Mt. Hunter; this study), McCall Glacier (Klein et al. 2016), Aurora Peak (Tsushima 2015), Mt. Wrangell (Yasunari et al. 2007), Bona-Churchill (Porter et al. 2019), Mt. Logan (Fisher et al. 2008), Eclipse Icefield (Yalcin et al. 2007), and Mt. Waddington (Neff et al. 2012). The map was produced using MATLAB (R2019b).

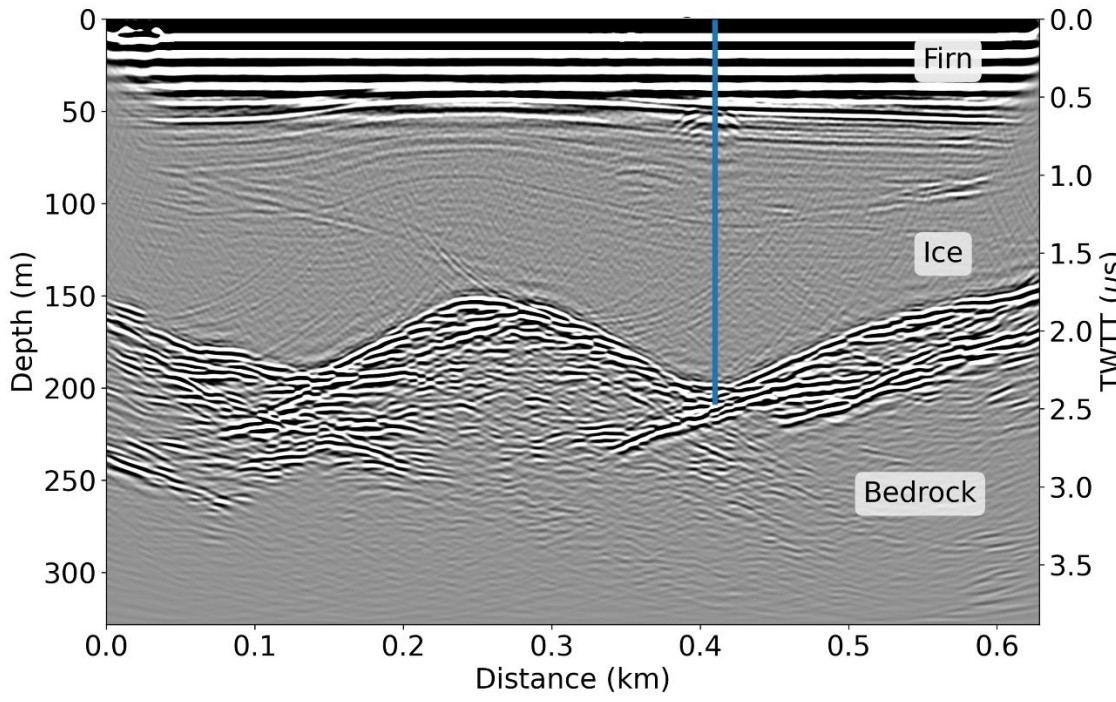


**Figure 2** Ground penetrating radar profile collected with 10 MHz BSI radar across the Begguya
plateau in 2022. Standard processing techniques were applied to the data using ImpDAR
(Lilien et al. 2020). The Two-Way Travel Time (TWTT) is plotted on the y-axis on the right
side. The Denali ice core drilling (DEN-13B) is indicated by the vertical blue line.

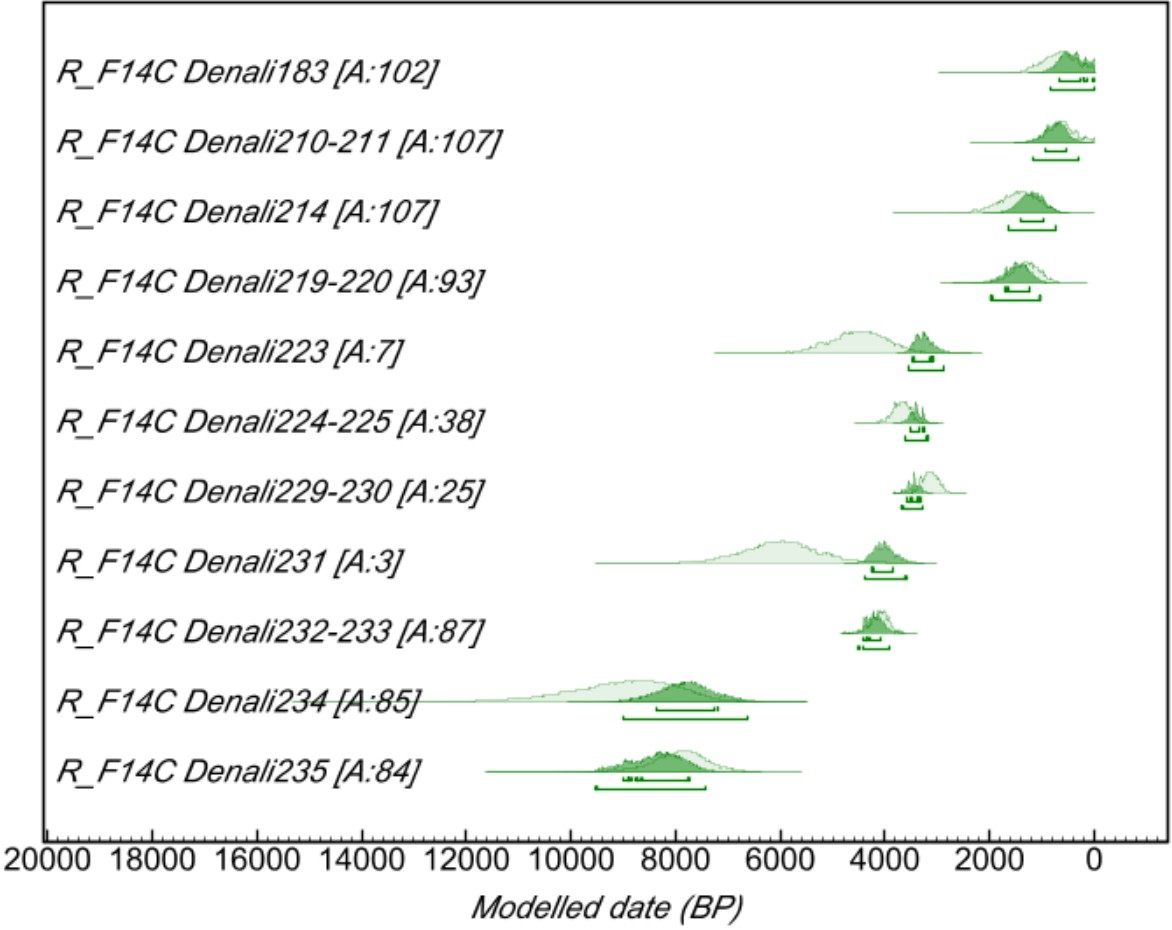


**Figure 3** Calibrated [14]C age probability distributions for samples from the Denali ice core (DEN-13B). as derived in OxCal v4.4.4 using the IntCal 20 radiocarbon calibration curve (Ramsey 2021, Reimer et al. 2020). Light green areas indicate the priori age probabilities, the dark green areas the posterior probabilities when sequential ordering of samples is assumed (see main text). The Agreement Index (*A*) indicates overlap between these two probability functions. *A* value < 60 indicates poor agreement (see main text). The 1σ and 2σ range is indicated by the lines below the probability distribution areas.

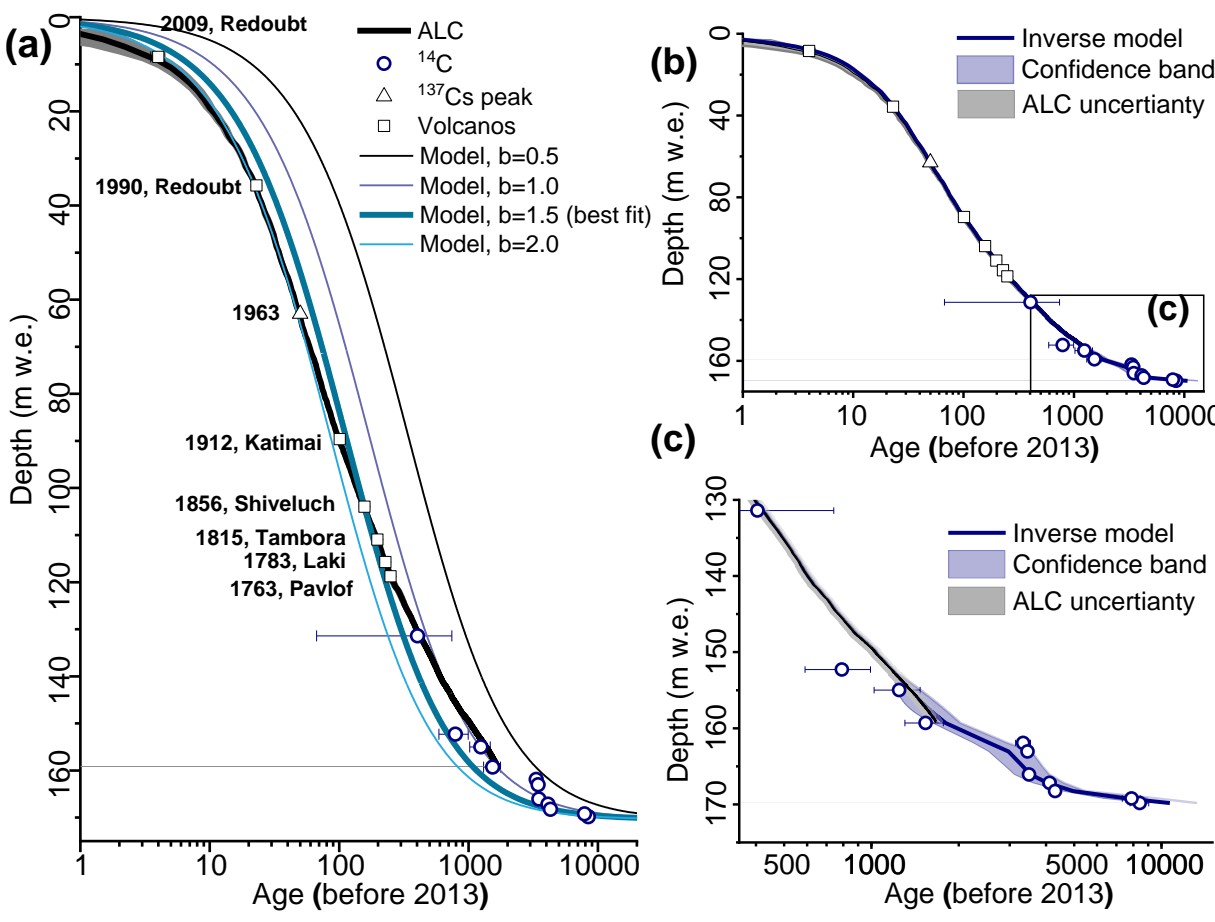

824

**Figure 4** Denali ice core (DEN-13B): annual layer counting (ALC), dating horizons ($^{14}$C,
Volcanos, $^{137}$Cs peak) and modeled, continuous age-depth relationship (1D ice flow model, see
main text). (a) Model output for constant accumulation rates ($b$, in m w.e. yr$^{-1}$). (b) Modeled
age-depth relationship for variable $b$ (inverse model). (c) Zoom of the deepest part. All error
bars indicate the 1σ uncertainty.

830

831

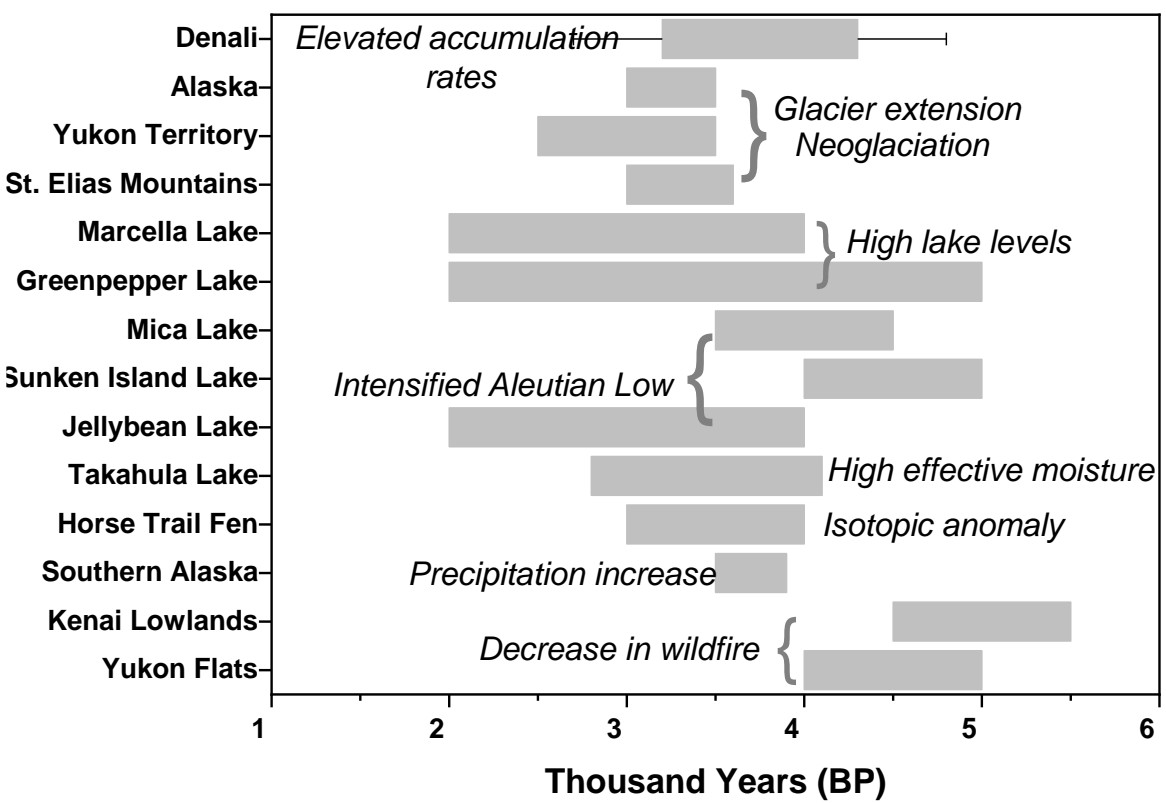

832

**Figure 5** Regional paleoclimate changes as reported in previous studies (Anderson et al. 2005a, 2005b, 2016, 2019, Anderson et al. 2006, Broadman et al. 2020, Clegg and Hu 2010, Denton and Karlén 1977, Heusser et al. 1985, Jones et al. 2019, Kelly et al. 2013, Schiff et al. 2009, Solomina et al. 2015) and the period of elevated annual net accumulation rates indicated in the Denali ice core DEN-13B (this study, see main text).

838

839

840