# Peer review of "Early Holocene ice on the Begguya plateau (Mt. Hunter, Alaska) revealed"

_The Cryosphere, 2023_

## Author Comment (AC2)

Reviewer #1

This paper presents the first reliable record of early Holocene ice in the Alaskan high mountains, where many scientists have attempted to locate it for many years. The finding of this study have the potential to inspire further exploration of Alaskan glaciers as a valuable paleoclimate proxy. Therefore, I strongly recommend the editor to accept this manuscript for publication in The Cryosphere. The manuscript is well-written, and the figures and tables are presented accurately. However, I would like to suggest the inclusion of several additional references that the authors may have overlooked.

Sasaki, H., Matoba, S., Shiraiwa, T. and Benson, C.S., Temporal variation in iron flux deposition onto the Northern North Pacific reconstructed from an ice core drilled at Mount Wrangell, Alaska, SOLA, 2016, 12, 287-290. DOI:10.2151/sola.2016-056.

https://www.jstage.jst.go.jp/article/sola/12/0/12_2016-056/_pdf/-char/ja

Shiraiwa, T., Goto-Azuma, K., Matoba, S., Yamasaki, T., Segawa, T., Kanamori, S., Matsuoka, K. and Fujii, Y., Ice core drilling at King Col, Mount Logan 2002, Bulletin of Glaciological Research, 2003, 20, 57-63.

https://web.seppyo.org/bgr/pdf/20/BGR20P57.pdf

We are grateful for the reviewer's comment. As suggested, both of these studies have been incorporated into the revised manuscript (section 4.2).

Reviewer #2

This manuscript presents the 14C ages of the a 210-meter long ice core drilled to the bottom from the summit plateau of Begguya (Mt. Hunter; Denali National Park, Central Alaska). The authors conclued that the basal ice on Begguya is at least of early Holocene origin, thus providing a potential paleo archive of Holocene climate in the North Pacific region. The results are reliable and the manuscript is well written.

We appreciated the reviewer's constructive suggestions. In response to the specific comments, we will provide detailed point-by-point replies below:

There are a few concerns that the authors may need to consider when revising the manuscript.

(1) In Figure 1, some of the ages below the ice core drilling sites may be misleading. For instance, the Aurora Peak ice core is with a length of 180.17m, while the age of 1734 CE corresponds to the depth of 149.68m w.eq. The authors need to verify if this age of 1734 CE refers to the basal ice? if not, how far the ice corresponding to 1734 CE is above the bottom? The same kind of verification is necessary for all the other ice cores shown in Figure 1.

We agree that the initial version of Figure 1 could be somewhat misleading without additional depth information. We have made the necessary revision to Figure 1 in order to provide explicit clarification about age and depth.

[Figure]

(2) Lines 116-117. Please confirm if or not this work provides the first radiometrically dated high latitude Northern Hemisphere ice core chronology?

Yes, we are sure, at least based on our best knowledge, that this is the case. If the reviewer possesses further information, we certainly would consider to include that.
(3) sections 2.1 and 3.2. Parts of the contents in these two sections duplicate.

Thank you for spotting this. Any duplications will be deleted in the revised manuscript.

(4) The author may want to consider moving section 3.1 into section 2.

We prefer to keep section 3.1 as part of the result chapter since the presented radar image is based on new data, which provides some additional, new insight near the bed surface.

(5) Lines 252-254. Slightly?

The original sentence in the manuscript is: "The 1σ $^{14}$C age range for Denali210-211 at 189.5 m (152.3 m w.e.) is 527-930 a cal BP, which is slightly younger than the annual layer

counting derived age of 1020-1200 a BP, but still in agreement within the 2σ uncertainty (317-1174 a cal BP)."

The statistical difference between calibrated $^{14}C$ ages (conventionally provided as an age range; here the 1σ range) and annual layer counted age range (based on the estimated counting uncertainty) is $380 \pm 220$ a. The statistically significant difference on the 1σ level is thus in the order of 20%. We therefore chose the word "slightly" here.

In the revised version of the manuscript, the sentence will be changed to:
"The 1σ $^{14}C$ age range for Denali210-211 at 189.5 m (152.3 m w.e.) is 527-930 a cal BP and with a possible age of 930 a cal BP only slightly younger than the annual layer counting derived age range of 1020-1200 a BP (in agreement within the 2σ range of 317-1174 a cal BP)."

Reviewer #3

General Comments:

This study provides a new radiocarbon-based chronology of an ice core obtained on the Begguya Plateau of Mt. Hunter (2013), located in the Alaska Range. As an application of recently developed radiocarbon dating methods for water-insoluble organic carbon (WIOC) and dissolved organic carbon (DOC), the study is successful in obtaining carbon samples of sufficient size to generate calibrated 14C age probability distributions that are largely in chronostratigraphic order. Further validation of the radiocarbon-based age model is achieved by comparisons with annual layer counting methods and inverse modeling for variable accumulation rates. Strong presentation and interpretation of these methods and results make a compelling argument that the authors have achieved continuous chronological constraint on the ice core through ~4000 to 5000 cal yr BP, or down to ~168 m depth for the ~209 m long core. For the two ages obtained below that depth, and for the author's interpretation of their significance, there are some additional questions to be addressed (see below). Although the authors provide possible implications for regional Holocene hydroclimate with some good hints shown by a bar diagram in Figure 5, this presentation over-generalizes the depth and breadth of the datasets and in some instances mis-characterizes the aspects of hydroclimate that they represent. Since precipitation records provided by high elevation ice cores combined with lower elevation records are needed to better understand the complex climate history of the Gulf of Alaska region, hopefully there are plans for more thought and exploration on these aspects to come.

Thanks for the thoughtful inputs and constructive suggestions. The main focus of this manuscript is on the new radiocarbon results and the chronology for the Denali ice core. Therefore, the section about regional Holocene hydroclimate is intended to be general, and findings are formulated as a hypothesis and not as finalized conclusions. We absolutely plan "for more thought and exploration on these aspects to come" as the reviewer suggests. In any case, the reviewer provided highly appreciated additional

input and suggestions which will be beneficial to this manuscript, and which we will happily incorporate into a revised version. Please find point-by-point replies to the detailed comments from the reviewer below, where we will also try to answer the reviewers questions about the ages obtained for the deepest section (below ~168 m).

Specific Comments:

> To allow detailed answers, the below comments by the reviewer were in cases split into smaller sections.

1.Presentation of the two lowermost ages is currently not sufficient to conclusively support the interpretations and proposed implications.

We respectfully disagree with the reviewer's view here (see details below).

A prominent feature of the chronostratigraphic results is the abrupt change in ice accumulation rates between samples 232-233 and 234 (Table 1), as shown in Figure 3 (and the age models in Figure 4, although the horizontal log scale of age somewhat masks its appearance).

There is no abrupt change in ice accumulation rates between samples 232-233 and 234. Please also note that no accumulation rates are presented in Table 1. Figure 3 shows the ages but not on a depth scale and Figure 4 shows the age-depth relationship from which changes in accumulation rates can be reconstructed with the help of an ice flow model. The reconstructed accumulation rates are shown in Supplementary Figure S2 and described in the main manuscript (see section 4.1.).

An abrupt change in accumulation rates is observed for (i) the most recent time period starting around 1750 CE over a period of around 250 years, a section dated by annual layer counting (increasing), (ii) rather abrupt over a period of around 500 years between ~3.2 and ~2.7 ka BP (decreasing; between samples 223 and 224-225/229-230) and (iii) a smaller change, roughly half compared to (i) and (ii), at around 4.3 ka BP (increasing; between samples 231 and 232-233).

We think that the reviewer was misreading thinning of annual layers to be caused by changes in accumulation rates. While this may typically be the case for other archives such as lake sediments, this is not the case for ice core archives. For those, especially for cores extracted from cold glacier sites where the ice is frozen to bedrock (no basal sliding) and particularly for alpine glaciers, layer thinning, exponentially increasing with depth, is related to the physical properties and the flow dynamics of ice, causing horizontal shear and thus physical thinning of layers in the archive itself. This is the main reason why dating of such cores is so challenging and absolute dates from radiometric methods such as radiocarbon dating is crucial and necessary.

For clarification, providing some useful background to readers from outside the ice core community, we will change lines 243-246 to:
"Calibrated [14]C ages range from 0.3 ± 0.3 ka cal BP at 166.2 m (131.4 m w.e.) depth to 8.4 ± 0.6 ka cal BP for the deepest sample (Denali235; 209.1 m or 169.8 m w.e.), the last sample above bedrock (0.6 m). These results show the characteristic exponential increase in age with depth, expected for a cold glacier archive due to the associated ice flow dynamics (e.g. Dansgaard and Johnsen, 1969, also see section 4.1.), and most importantly reveal the ice of early Holocene origin in the Denali ice core (Table 1 and Figure 3)."

Ages from both depths are from WIOC, but the sample sizes are significantly different (54.8mg and 9.8mg, for depths of 206.6m to 208.1m and 208.1m to 208.8m, respectively). Notably the value of 9.8mg is just shy of the recommended 10mg threshold described in lines 239-242. Yet, it must be asked why this sample was considered acceptable when, for example, a value of 9.2mg for sample 209 (187.8-188.7 m), which provided an age that is stratigraphically 'too old' compared to those above and below it, was not included in the chronostratigraphic analyses. It would seem very possible that the age of 6.2-9.6 cal yr BP provided by the 234 sample is also stratigraphically too old in a similar manner. Additional discussion is needed to explain why this sample was included.

First, the conclusion of the reviewer that sample 209 is stratigraphically too old is not quite right. This sample, without question, has quite a large uncertainty (i.e. the range of possible ages is wide). Please be aware, that radiocarbon ages are conventionally reported as a range, with any age in this range being a likely solution of the "true" age (note that the probability distribution is not of Gaussian shape because of the "wiggles" in the radiocarbon calibration curve). If considering the entire range, or the middle of the range (if assuming a Gaussian distribution), then yes, the result of sample 209 with a one sigma age range from 977-1925 years (see Table S2) appears to be on the upper end, compared with the annual layer counting providing a possible age range from 980-1090 years. But obviously, [14]C dating for that section yielded probable ages which are in agreement with the independent dating by annual layer counting. Further, while the [14]C age for the sample above was dated to be younger, which is to be expected (increasing age with depth), a younger age was indeed derived for the sample below (Denali210-211). Sample Denali210-211, was discussed in lines 251-254, because that sample, again compared to the annual layer, appears to be slightly too young in age (and of course, thus younger also than sample 209). For these reasons, we argue that the conclusion that sample 209 "provided an age that is stratigraphically 'too old' compared to those above and below it" is incorrect, both statistically and in the overall context.

Nevertheless, we will answer/discuss the actual question (data selection) and the concern of the reviewer (possibility of sample 234 providing a too old age) in the following:
*Data selection*

We see the reviewers concern but would like to point out, that we believe the selection of data is in the end irrelevant for the main conclusion of the manuscript. This is, because even if considering all samples as presented in the supplementary material (Table S1), the main conclusion of the manuscript that the "… very bottom ice…" of the Denali ice core is "…of early Holocene age" would not change. However, realizing that the criteria for accepting or rejecting samples was maybe not as clearly described in the current manuscript version as we believed, we provide an in-depth explanation here and will make some minor adaptations in a revised manuscript version (see further down):

The reviewer is correct that sample size is important. Reviewer 4 raised a similar question regarding the exclusion of sample 215-216, which is in fact a great example to demonstrate the risk of inconsistency when selecting or rejecting samples based on a subjective view. In order to avoid such subjective picking of samples, we strictly applied the recommended threshold of 10 µg C (see Uglietti et al., 2016), excluding samples for which "…the total carbon amount … was significantly below this 10 µg C" (see lines 237-240; Note, that for transparency all values are provided in the Supplementary anyhow; referred to in lines 240-242). Thereby, we did not care if a sample, as the reviewer put it: "provided an age that is…" (…seems… would be the better word here) "…stratigraphically 'too old' compared to those above and below it ". The carbon mass for Denali209 and Denali234 was 9.2±0.7 µg and 9.8±0.7 µg, respectively (8.8±0.6 µg for sample 215-216). We realize though, to avoid confusion, we should have been rounding the sample C mass provided in Table 1 and Supplementary Table S2 to the relevant decimal only (or at least provide the associated uncertainty as well). We will adapt this accordingly in the revised version.

Here, for completeness, we will provide some additional background but these details could also be found in the literature cited in the manuscript. The theoretical dating uncertainty for the applied method can be calculated, showing to be sharply increasing below 10 µg C for samples older than around 1000 years (the uncertainty is higher for samples of <500 years due to a "plateau" in the [14]C calibration curve during that more recent period). The threshold of 10 µg C, recommended in Uglietti et al. (2016), is based on these theoretical calculations. It was set to a value to ensure acceptable dating uncertainties in the order of 20% or below for any sample older than around 1000 years. Note that the uncertainty also depends on the age of the sample, becoming smaller with an increase in age (note: for sample 234, relatively old in age, the small size of just 10 µg C is thus generally less of a concern). We here visualize the theoretical calculations of the uncertainty from Uglietti et al. (2016) (y-axis) in dependence of the sample size (x-axis) and for different ages of a hypothetical sample (line coloring according to the legend). See figure below (u = uncertainty; lower panel: [14]C age in years BP is the conventional radiocarbon age while [14]C age in years calBP is the calibrated radiocarbon age).

[Figure]

*Age of sample 234*

Certainly, there is some probability, that the derived age for sample 334 might be slightly too old (or also probable, too young). This is pure statistics. Any measurement is associated with a measurement precision (regarding the accuracy of the method, please see Uglietti et al., 2016 and Fang et al., 2021). Dependent on that precision, the measured data is expected to scatter around the mean value and roughly, the 1σ range of an individual measurement should include the mean value in around 7 out of 10 cases. We think that the question the reviewer should rather be interested in is, if this obvious fact is relevant for the main conclusion of the manuscript (ice of early Holocene origin)? To answer this, please note, that the final chronology presented in our study does not rely on individual data points at any depth. This increased the confidence in the final age scale, allowing us to come to the firm conclusion about the finding of early Holocene ice (note that the achieved precision, i.e. the scatter/noise in the data defined the uncertainty of that final scale). See also, and find more details in, our response later on.

For clarity in the revised version of the manuscript, the according part of section 3.3. will be reformulated as follows:

"Samples containing less than 10 µg carbon are generally characterized by a wide age range. A reduction in the dating precision - and because of the non-constant atmospheric [14]C content over time, the need to consider all probable solutions for a final calibrated [14]C age, by convention thus reported as a range, to be as likely - is expected due to the small carbon amount available for analysis. Small amounts cause reduced AMS measurement precision and additionally, because of a lower and thus unfavorable signal-to-noise ratio (i.e. the ratio between size of sample and procedural blank), a

general increase in the overall analytical uncertainty, finally translating into the observed wider range of possible age solutions. Although we used a considerable amount of ice for each sample (~1 kg), the total carbon amount in 5 samples was significantly below this 10 µg C threshold recommended to obtain a reliable dating with a final uncertainty < 20% for samples older than around 1000 years (Uglietti et al., 2016). These samples will thus not be discussed in the following (but can be found in the supplement material, together with calibration results without sequence constraint).

Uglietti, C., A. Zapf, T. M. Jenk, M. Sigl, S. Szidat, G. Salazar and M. Schwikowski, Radiocarbon dating of glacier ice: overview, optimisation, validation and potential, The Cryosphere, 2016, 10(6), 3091-3105. DOI: 10.5194/tc-10-3091-2016.

If it is a correct age, then the implication is that up to ~4000 years of record is compressed into 1-2 m of ice across this interval. This begs additional questions about the englacial stratigraphy for which lines 197-198 say there is no conclusive evidence for stratigraphic continuity or discontinuity. If the age data is accepted, then additional investigation-discussion would be helpful to explore either additional evidence, for example related to melt layers or changes in precipitation seasonality, which provide possible explanations for the rapid reduction in accumulation and/or preservation of ice in the core at this time.

> ➢ For context, because we had to split the comment in smaller sections, the reviewer here refers to the ages in the deepest sections of the core.

Quite the opposite: The fact that most of the record in terms of time is compressed into a (very) short depth interval at the bottom of the glacier (i.e. ice core) is exactly what is expected for a cold (ice frozen to bedrock with no basal sliding) glacier archive. In other words, the exponential increase of age with depth is expected (characteristic) and questions would need to be asked if this would not have been reflected in our results. For additional details, see our reply to the comment regarding layer thinning further up or the related literature cited in the manuscript.

The quote of lines 197-198 made here by the reviewer lacks the context provided in the manuscript. There is no statistical evidence for stratigraphic discontinuity based on the $^{14}C$ dating results. There is also no evidence for discontinuity visible in the ice penetrating radar data. However, because this data is unfortunately not very clean at the bottom due to reflections and scatter from the nearby bed surface this data can also not provide clear evidence of continuity.

As already explained in a reply above, our results do not show a "…rapid reduction in accumulation and/or preservation of ice in the core at this time…" (see Supplementary Figure S2). Anyhow, melt layers and changes in seasonal precipitation are not an explanation for the strong, exponential thinning of annual layers with increasing depth in cold glacier archives, which is in fact not only very typical but even expected for

cold glaciers due to the physical properties and flow dynamics of ice. For examples from alpine glaciers e.g. see Figure 6 in Uglietti et al. (2016) or for an example from the Greenlandic ice sheet e.g. see Figure 5 in Dansgaard and Johnsen (1969).

The lowermost age, sample 235 at 208.8-209.4 m depth, was the only age from the core obtained from DOC. Although it does not suffer from sample size issue as the WIOC age above, it does raise the question about the propensity for DOC ages to be stratigraphically "too old" due to old carbon sources. One way to address this concern would be to date a few other intervals down the core with both DOC and WIOC to establish that the two materials can be confidently compared in this core. Due to the large sample sizes needed for the analyses, it is understandable that this may not be readily feasible. However, without such a data comparison, it would strengthen the quality of the discussion to include an acknowledgement of this possibility and add a more parsimonious evaluation of the implications.

In summary, there is a reasonable possibility that the lowermost two ages are stratigraphically "too old" due to analytical and source carbon reasons and the manuscript could be improved to account for and acknowledge this possibility with corresponding adjustments to the certainty for which the implications are presented. For example, lines 440-442 that states the results 'clearly indicates' ice of early Holocene age are rather more suggestive and therefore to be developed further.

Previous reported [14]C age from DOC in ice indicate that [14]C-DOC age has never been observed to potentially suffer from old carbonate sources. On the contrary, observed was a potential for a bias towards too young ages. See May et al., 2013 and Fang et al., 2021.

The suggestion by the reviewer to compare results from DOC and WIOC on parallel samples is a good idea. In fact, such comparison of [14]C dating from the DOC or WIOC fraction was performed and can be found in Fang et al., 2021. There, results from multiple samples for each of 4 different ice cores extracted from glaciers on globally distributed regions were compared. No significant difference in age was observed between these samples, except for one case where both (i) mineral dust content in the ice was extremely high (Tibetean Plateau with high input of wind blown dust from the nearby Taklamakan desert), resulting in a bias to older ages by a few hundred years for WIOC due to incompletely removed carbonates, and (ii) the site was very high (6010 m asl.) and accumulation rates were low (around 0.14 m w.e.) resulting in a detectable DOC offset to younger ages of up to 2000 years due to in-situ [14]C production in the ice . Both, (i) and (ii) is less of a concern for the site of this study. In any case, the DOC [14]C in-situ production can be estimated by calculations (see Fang et al., 2021) and was here considered with the correction accounting to plus 300±200 years, with the upper estimate similar in size as the analytical precision (see Supplement Table S1 and our response to a related comment from Reviewer 4).

In summary, we (a) do have results from two different samples close to the bedrock,

each measured independently and on a different OC fraction, both indicating ice of early Holocene age, i.e. older than ~5 ka (sample 234 with a probability of >95% and the sample 235 below with a probability >99%). Further, (b) age estimates based on an ice flow model considering the principals of ice flow physics yields early Holocene ages at the bottom for assumed steady-state conditions and different values of constant accumulation (see Figure 4a). And finally (c)[*], an inverse modeling approach which combines and benefits from (a) and (b), not relying on individual data points but the entire data set with full error propagation from analytical data to final results being carried out (see lines 343-346), which clearly results in early Holocene ages at the bottom of the core. For these reasons, we are confident with close to 100% certainty about our statement "of early Holocene ice" at the drill site.

[*]In (c), any scatter in the dating results is accounted for, and ultimately expressed by the size of the final dating uncertainty of the final model derived chronology (the bigger the scatter, the bigger the final uncertainty). More information about the relation/dependence between analytical precision and scatter in experimental data can easily be found (basic statistics).

May, B., Wagenbach, D., Hoffmann, H., Legrand, M., Preunkert, S., and Steier, P.: Constraints on the major sources of dissolved organic carbon in Alpine ice cores from radiocarbon analysis over the bomb-peak period, J. Geophys. Res.-Atmos., 118, 3319–3327, https://doi.org/10.1002/jgrd.50200, 2013.

Fang, L., Jenk, T. M., Singer, T., Hou, S., & Schwikowski, M. (2021). Radiocarbon dating of alpine ice cores with the dissolved organic carbon (DOC) fraction. The Cryosphere, 15(3), 1537-1550.

A reconsideration of the statement that the results 'confirm that at least some glacier ice… survived… the Holocene thermal maximum' (lines 397-398) could include addressing if there is any evidence to doubt that glacier ice at 3900 m elevation persisted during the early Holocene.

We agree that given the low mean annual temperature (-17° C) of the site, there is very little possibility of this glacier having disappeared during the mid-early Holocene.   We have removed this statement from the revised version.

Furthermore, the stated conclusion also raises questions about how basal ages of mountain glaciers serve to provide evidence for ice presence (or not) when they may be in constant downward motion.

The Denali ice core site is a cold glacier site with the ice being frozen to the bedrock where it thus persists over time (not flowing away). The combination of the downward flow of ice in the upper part (constantly fed from the accumulation/deposition upstream) and being frozen to bed at bedrock is causing strong horizontal shear which causes the characteristic exponential thinning of layers with depth (see also comments above).

2. Regional paleoclimate

The primary subject of this paper is the radiocarbon data, and the context is regional paleoclimate. So, although it is not a focus or objective to deeply survey the regional paleoclimatic data, there are some important general aspects that could be better clarified.

First, while the notion of a "early Holocene thermal maximum" may have meaning for the cryosphere on a pan-Arctic scale, in Alaska a thermal expression is vague, if at all, by low elevation terrestrial proxies, including pollen. Rather, corresponding changes in effective moisture are more commonly reflected by the proxy records.

Prominent oxygen isotope studies document changes in atmospheric circulation patterns related to the strength and position of the Aleutian Low (Jellybean with the PRCol record from Logan summarized by Anderson et al., 2016, Takahula and more recently Tangled Up in the Brooks Range by Anderson et al., 2018: these records should not be primarily characterized as 'lake level' such as in Figure 5).

Subsequent studies show how those changes in atmospheric circulation may be related to changes in effective moisture on the leeward sides of the major mountain barriers, ie., lake level (Marcella, Seven Mile Lake by Anderson et al., 2011, and more recently Track Lake in the Yukon Flats by Anderson et al. 2018 and Squanga Lake in the interior Yukon by Lasher et al., 2021).

However, much like the complexities discovered when comparing the ice cores from Logan with Hunter by Osterberg et al., 2017, it may not always be a correct assumption that the hydroclimate of the SW Yukon and Alaska Range are always the same.

Most closely located to Hunter is a carbon isotope record from Dune Lake, on the north (leeward) slope of the Alaska Range (Finney et al., 2012), which suggests that late Holocene changes to winter precipitation seasonality has played a role in lake level through groundwater, which could in theory be related to Alaska Range glacial ice. For the windward side of the range, on the Kenai Peninsula, there is currently the Horse Trail Fen peat oxygen isotope record (Jones et al., 2019) with additional work on lake isotope records currently underway in Upper Cook Inlet (IALIPA abstract by Anderson et al., 2022).

In summary, while there is a rich and growing record of the hydroclimatic changes in Alaska during the Holocene, many questions remain about the role of temperature, moisture and precipitation seasonality in relation to the complex topography of the region.

With an improved characterization of the state of knowledge, the contributions made by this study of the Holocene ice accumulation on Mt. Hunter will be more clear. This paper has begun to provide the needed chronology for additional contributions, and it is hoped that more in depth comparative analyses are planned.

Thank you very much for this helpful comment. We absolutely agree with you that there is a rich and growing literature about Holocene hydroclimate changes in the Alaska region and that the interpretation of these changes must involve the collective

analysis of multiple sites in order to interpret the complexities of seasonality, rain shadows, temperature and circulation dynamics. We are not attempting to do all of that here, especially since Anderson et al. (2016) and Kaufman et al. (2016) provided excellent reviews recently. However, we have significantly expanded the discussion of regional paleo-hydroclimate studies in section 4.3 to make it clear that our results are consistent with the prevailing idea of enhanced precipitation (at least at windward sites) around 4k BP and that this is likely associated with stronger/deeper Aleutian Low. Given our uncertainties, we are very cautious about making strong inferences here about accumulation rates in the mid-Holocene, but our hope is that the new revision makes it clear that our results are consistent with and supporting of previous work. We have also changed Figure 5 to characterize the Jellybean Lake record as "intensified Aleutian Low" and the Takahula Lake record as "increased effective moisture" to better reflect the conclusions presented in Anderson et al. (2005) and Clegg and Hu (2010).

[Figure]

Reviewer #4

Fang et al. present new 14C ages measured on WIOC and DOC from an ice core from the Begguya plateau. I think this is a very interesting study that explains well the new measurements and results. I have just a few minor comments that the authors should consider in an updated version of the manuscript.

We are grateful for the suggestions. For the concerns that the reviewer raised, please see our point-by-point reply below:

I am wondering about the modelling approach with the variable accumulation rate (figure S2). I understand the approach the authors did but less thinning at the bottom would directly lead to lower accumulation rates. How realistic is the assumption of constant thinning and are the uncertainty bands in the accumulation rate (figure S2) realistic?

We do not quite understand why the reviewer thinks there is "less thinning at the bottom". Thinning of annual layers is exponentially increasing with depth, which is not only particularly characteristic for cold alpine glacier archives (ice frozen to bedrock and thus no basal sliding) but also the case for polar ice archives. Also, less "thinning" (in essence meaning thicker layers) would not "directly lead to lower accumulation rates". On the contrary, instead this might rather indicate higher accumulation rates.

The thinning is not constant but exponential. Only the thinning parameter is kept constant (which is in the exponent of the model function). In any case, for varying accumulation, also the thinning rate will vary since accumulation rate and strain rate are dependent (although accumulation rate is only a minor factor in the determination of the strain rate, i.e. not in the exponent of the function). For details, please see Bolzan, 1985 or consult Uglietti et al. (2016) where the relevant equations are summarized and reformulated for calculation of layer thickness and strain rate as a function of depth.

The confidence bands in Figure S2 were statistically derived and include full propagation of errors (including analytical uncertainties etc.). Some assumptions such as steady state ice thickness and temperature were made for the modeling (and as a consequence the reconstruction). For future studies, focusing specifically on accumulation (not the main topic of this study), improvement by more sophisticated modeling approaches can likely be achieved, but currently we have no reason to expect changes exceeding the confidence band shown here.

Bolzan, J. F., Ice flow at the Dome C ice divide based on a deep temperature profile, Journal of Geophysical Research: Atmospheres, 1985, 90(D5), 8111-8124.

Uglietti, C., A. Zapf, T. M. Jenk, M. Sigl, S. Szidat, G. Salazar and M. Schwikowski, Radiocarbon dating of glacier ice: overview, optimisation, validation and potential, The Cryosphere, 2016, 10(6), 3091-3105. DOI: 10.5194/tc-10-3091-2016.

It would be nice to have an explanation why sample "Denali215-216" was excluded but sample "Denali234" not. The latter sample is very important for the age model but it has a lower WIOC content than "Denali215-216". I see the reason for excluding "Denali215-216" as it does not fit into the 14C age sequence but I am wondering if this questions the robustness of the "Denali234" result.

The carbon mass of Denali215-216 and Denali234 were 8.8 µg C and 9.8 µg C, respectively. The threshold of 10 µg C to reject smaller samples, recommended in a

previous study by Uglietti et al. (2016), was strictly applied in order to avoid subjective "picking" of results. Find more details in our response to a related comment from Reviewer 3.

Why was the "in-situ" correction done with an accumulation rate that is lower than the reconstructions based on the modelling in this study (Figure S2 lower panel)? The authors write that they take the average accumulation rate from Winski et al., (2017) but figure S2 shows lower values for the Winski model. Please explain.

This is an interesting question. The reconstructed accumulation rates require the input of results from the $^{14}$C dating (for model constraint). Thus, the reconstructed value is initially unknown. For this reason, a best value had to be estimated first. We chose a value of 1.0±0.5 m w.e., based on the derived annual values by Winski et al. 2017, ranging from 0.2 to 2.0 m w.e. for the time period 810 to 2013 CE. The size of correction in terms of age, resulted in plus 300±200 years with the uncertainty, like all others, in the following always being fully propagated. With even the upper estimate (for 0.5 m w.e.) not exceeding the achieved dating precision defined by the analytical uncertainty (see Table S1 in the Supplementary), this correction does not seem crucial overall. Also, the value finally reconstructed for the section of this sample (and defined also by the two samples above) was 1.3±0.4 m w.e., fairly close to the estimated and used value.

In the revised version of the manuscript, the according section will be reformulated to: "F$^{14}$C of DOC was corrected for contribution from $^{14}$C in-situ production following Fang et al. (2021). The applied small shift in F$^{14}$C of 0.019±0.010 was derived using an in-situ production rate of 260.9 $^{14}$C atoms $g_{ice}^{-1}$ $a^{-1}$ as a best estimate for the site latitude and elevation (Lal et al. 1987, Lal and Jull 1990, Lal 1992), an average accumulation rate of 1.0 ± 0.5 m w.e. (a best initial guess based on the derived annual values by Winski et al. 2017, ranging from 0.2 to 2.0 m w.e. for the time period 810 to 2013 CE), and assuming an average incorporation into DOC of 18±7% (Hoffmann, 2016). This correction shifts the calibrated age by plus 300±200 years with the uncertainty, like all others, in the following always being fully propagated and noting that the upper estimate does not exceed the achieved dating precision defined by the analytical uncertainty (see Table S1 in the Supplementary). For all samples, final calibrated radiocarbon ages were derived by calibrating final F$^{14}$C values using OxCal v4.4.4 (Ramsey 2021) with IntCal20 (the Northern Hemisphere calibration curve; Reimer et al. 2020) and the OxCal in-built sequence model (Bayesian approach-based deposition model; Ramsey 2008, Ramsey 2017)."

In the conclusion the authors write: "These were achieved by a slight adaptation of the WIOC 14C-dating method, allowing for larger ice samples of up to around 1 kg of ice and the use of a new technique for 14C dating of the DOC fraction. From the text it was not clear to me what exactly this "adaptation" was. Similarly, it would be good to write clearly what the "new technique" for DOC measurements includes in comparison to previous measurements?

The routine for WIOC analysis is to melt the ice sample in pre-cleaned PETG jars (1L, Semadeni) prior to filtration. In order to analysis ice samples >1kg, a splitting of the sample for melting was required and the overall filtration time had to be increased. Using artificial ice produced from ultra-pure water, the procedures had to be optimized to ensure low blanks. This is what we referred to with "adaptation".

In the method section (2.2) after line 143, we will add the following sentences (and make some additional changes to the wording for better language):
"In order to process such large sample volumes, a splitting of the sample for melting was required and the overall filtration time had to be increased. Using artificial ice produced from ultra-pure water, the adapted procedures were tested to reach low blanks similar to the ones previously achieved for smaller samples (Jenk et al., 2009; Uglietti et al., 2016; Fang et al., 2019). Otherwise, the samples for WIOC 14C-daing were prepared following the protocol described in Uglietti et al. (2016) with a brief summary provided here." and in line 146-147:
"After melting of the sample in a pre-cleaned jar (1L, PETG, Semadeni), due to the size split in two, the carbonaceous particles contained as impurities in the sample ice were filtered onto a prebaked quartz fiber filter (Pallflex Tissueqtz-2500QAT-UP)."

In the Conclusion the sentence referred to by the reviewer will be changed to:
"These were achieved by small adaptations in the ice sample preparation procedures for the WIOC $^{14}$C-dating method which allowed processing of larger ice samples up to >1 kg of ice, and the application of a new technique for $^{14}$C dating of the DOC fraction, which benefits from higher concentration levels in ice compared to the WIOC fraction (by around a factor of three)."

The DOC technique was already described in lines 160-165 with a reference to the according method paper (Ling et al., 2019): "For the deepest sample from ~209 m depth (Denali 235) the available amount of ice was very limited (~200 g). To ensure sufficient mass of carbon for final AMS analysis, the $^{14}$C dating was performed on the DOC fraction for which a higher concentration compared to the WIOC fraction is expected (Legrand et al. 2013). By a catalyzed UV-Oxidation in a dedicated system, DOC was converted to $CO_2$ which was then cryogenically trapped and flame sealed in glass ampules for final AMS analysis (Fang et al. 2019)."

In the revised version the following change will be made: "…for final AMS analysis. Details can be found in Fang et al. (2019)."

The basis for preferring WIOC over DOC 14C measurements could be very briefly mentioned for the non-experts in the method.

We do not think there is a preference for WIOC over DOC and are also not aware of any study concluding this. Certainly, we did not prefer WIOC over DOC in this study.

However, $^{14}$C dating with the WIOC fraction has been a well-established method at the time the Denali samples were processed and measured (in 2018 and 2019), which was not (yet) the case for the DOC method. As we mention in this study here, the concentration of DOC in the ice is higher compared to WIOC, which can be beneficial especially for sites with generally low organic content in the ice (e.g. more polar regions).